# Prevalence of Germline Variants in a Large Cohort of Japanese Patients with Pheochromocytoma and/or Paraganglioma

**DOI:** 10.3390/cancers13164014

**Published:** 2021-08-09

**Authors:** Masato Yonamine, Koichiro Wasano, Yuichi Aita, Takehito Sugasawa, Katsutoshi Takahashi, Yasushi Kawakami, Hitoshi Shimano, Hiroyuki Nishiyama, Hisato Hara, Mitsuhide Naruse, Takahiro Okamoto, Tadashi Matsuda, Shinji Kosugi, Kazuhiko Horiguchi, Akiyo Tanabe, Atsushi Watanabe, Noriko Kimura, Eijiro Nakamura, Akihiro Sakurai, Kiyoto Shiga, Kazuhiro Takekoshi

**Affiliations:** 1Laboratory of Laboratory/Sports Medicine, Division of Clinical Medicine, Faculty of Medicine, University of Tsukuba, 1-1-1 Tennodai, Tsukuba 305-8577, Ibaraki, Japan; yonamine.masato.fu@u.tsukuba.ac.jp (M.Y.); take0716@krf.biglobe.ne.jp (T.S.); y-kawa@md.tsukuba.ac.jp (Y.K.); 2Division of Hearing and Balance Research, National Institute of Sensory Organs, National Hospital Organization Tokyo Medical Center, Tokyo 152-8902, Japan; wasano@a5.keio.jp; 3Department of Endocrinology and Metabolism, Faculty of Medicine, University of Tsukuba, 1-1-1 Tennodai, Tsukuba 305-8577, Ibaraki, Japan; yaita@md.tsukuba.ac.jp (Y.A.); hshimano@md.tsukuba.ac.jp (H.S.); 4Division of Metabolism, Showa General Hospital, 8-1-1 Hanakoganei, Kodaira 187-8510, Tokyo, Japan; ktaka-tky@umin.ac.jp; 5Department of Urology, Faculty of Medicine, University of Tsukuba Hospital, 2-1-1 Amakubo, Tsukuba 305-8576, Ibaraki, Japan; nishiuro@md.tsukuba.ac.jp; 6Department of Breast and Endocrine Surgery, Faculty of Medicine, University of Tsukuba, 1-1-1 Tennodai, Tsukuba 305-8577, Ibaraki, Japan; harahisa@md.tsukuba.ac.jp; 7Institute of Clinical Endocrinology and Metabolism, National Hospital Organization Kyoto Medical Center, 1-1 Fukakusamukaihatacho, Fushimi-ku 612-8555, Kyoto, Japan; m-naruse@takedahp.or.jp; 8Endocrine Center and Clinical Research Center, Ijinkai Takeda General Hospital, 28-1 Ishidamoriminamicho, Fushimi-ku 601-1495, Kyoto, Japan; 9Department of Breast and Endocrine Surgery, Tokyo Women’s Medical University, 8-1 Kawada-cho, Shin-juku-ku 162-8666, Tokyo, Japan; okamoto.takahiro@twmu.ac.jp; 10Department of Urology and Andrology, Kansai Medical University, 2-5-1 Shin-machi, Hirakata 573-1010, Osaka, Japan; matsudat@hirakata.kmu.ac.jp; 11Department of Medical Ethics/Medical Genetics, Kyoto University School of Public Health, Yoshidakonoecho, Sakyo-ku 606-8501, Kyoto, Japan; kosugi@kuhp.kyoto-u.ac.jp; 12Division of Endocrinology and Metabolism, Gunma University Graduate School of Medicine, 3-39-15 Showa-machi, Maebashi 371-8511, Gunma, Japan; k-hori@gunma-u.ac.jp; 13Department of Diabetes, Endocrinology and Metabolism, Center Hospital of the National Center for Global-Health and Medicine, 1-21-1 Toyama, Shinjuku-ku 162-8855, Tokyo, Japan; atanabe-endo@umin.ac.jp; 14Division of Clinical Genetics, Kanazawa University Hospital, 13-1 Takaramachi, Kanazawa 920-8641, Ishikawa, Japan; aw3703@staff.kanazawa-u.ac.jp; 15Department of Pathology, National Hospital Organization Hakodate National Hospital, 18-16 Kawahara-cho, Hakodate 041-8512, Hokkaido, Japan; kimura.noriko.sf@mail.hosp.go.jp; 16Department of Urology, National Cancer Center Hospital, 5-1-1 Tsukiji, Chuo City 104-0045, Tokyo, Japan; einakamu@ncc.go.jp; 17Department of Medical Genetics and Genomics, Sapporo Medical University School of Medicine, Minami 1-jo Nishi 17-chome Chuo-ku, Sapporo 060-8556, Hokkaido, Japan; sakurai.akihiro@sapmed.ac.jp; 18Department of Head and Neck Surgery, Iwate Medical University School of Medicine, 2-1-1 Idaidori, Yaha-ba-cho, Shiwa-gun 028-3695, Iwate, Japan; kshiga@iwate-med.ac.jp

**Keywords:** pheochromocytoma, paraganglioma, genetics, germline variants, Japanese, *SDHB*, *SDHD*, *VHL*

## Abstract

**Simple Summary:**

Pheochromocytoma/paraganglioma (PPGL) has been recognised as one of the most frequent inherited tumours with genetic heterogeneity based on studies in Caucasian populations. Early identification of germline variants is crucial for accurate treatment and follow-up in affected patients and relatives. However, there are only a few large cohort studies in Asia and none from the Japanese population. In this first comprehensive study of Japanese patients with PPGL, we found one in four PPGLs with apparently sporadic presentation harboured germline variant in any of the seven susceptibility genes (*MAX*, *SDHB*, *SDHC*, *SDHD*, *TMEM127*, *VHL*, and *RET*). *SDHB* was the most frequently mutated gene and was strongly associated with metastatic PPGLs. Our findings emphasise the importance of genetic testing in determining appropriate treatment and follow-up strategies for patients and relatives.

**Abstract:**

The high incidence of germline variants in pheochromocytoma and paraganglioma (PPGL) has been reported mainly in Europe, but not among Japanese populations in Asia. We aimed to study the prevalence of germline variants in Japanese PPGL patients and the genotype–phenotype correlation. We examined 370 PPGL probands, including 43 patients with family history and/or syndromic presentation and 327 patients with apparently sporadic (AS) presentation. Clinical data and blood samples were collected, and the seven major susceptibility genes (*MAX*, *SDHB*, *SDHC*, *SDHD*, *TMEM127*, *VHL*, and *RET*) were tested using Sanger sequencing. Overall, 120/370 (32.4%) patients had pathogenic or likely pathogenic variants, with 81/327 (24.8%) in AS presentation. *SDHB* was the most frequently mutated gene (57, 15.4%), followed by *SDHD* (27, 7.3%), and *VHL* (18, 4.9%). The incidence of metastatic PPGL was high in *SDHB* carriers (21/57, 36.8%). A few unique recurrent variants (*SDHB* c.137G>A and *SDHB* c.470delT) were detected in this Japanese cohort, highlighting ethnic differences. In summary, almost a quarter of patients with apparently sporadic PPGL in Japan harboured germline variants of the targeted genes. This study reinforces the recommendation in Western guidelines to perform genetic testing for PPGL and genotype-based clinical decision-making in the Japanese population.

## 1. Introduction

Pheochromocytoma (PCC) and paraganglioma (PGL) are neuroendocrine tumours derived from the chromaffin cells in the adrenal medulla and autonomic nervous system ganglia, respectively. Despite the anatomical distinction, PCC and PGL share a common pathological basis and genetic background and are collectively referred to as pheochromocytoma/paraganglioma (PPGL) [1,2]. PPGL can secrete catecholamines, the plasma or urinary metabolites which are essential for biochemical diagnosis [3], while excess catecholamines can cause cardio- or cerebrovascular complications. All PPGLs have the potential to metastasise to non-chromaffin tissues including bone, lung, liver, and lymph nodes; thus, the prefix term “benign” was abandoned for these tumours in the 2017 World Health Organization (WHO) classification of endocrine tumours [4].

PPGL is now considered to be the most frequent inherited tumour with genetic heterogeneity. Germline variants are present in around 30% of PPGLs [5,6,7,8]; more than 20 susceptibility genes have been identified over the last two decades [9,10]. Even in PPGL with apparently sporadic (AS) presentation (no familial/syndromic [FS] characteristics), pathogenic variants have been found with a frequency of 11–24% [11,12,13]. In this context, current guidelines recommend that germline genetic testing should be considered in all patients with PPGLs regardless of family history [2,14,15]. Identification of a predisposing germline variant enables risk assessment of distant metastases in probands and clinical surveillance of variant carriers in healthy relatives [1,2,9,16].

The genetic aetiology and genotype-phenotype relationship of PPGL have been studied extensively in Caucasian populations [5,6,11,17,18,19]. Recently, a large population study of 719 Chinese and 919 European patients demonstrated broad Sino-European differences in the genetic landscape and clinical presentation of PPGL [20]. This study suggested that the genetic background of PPGL and the associated genotype–phenotype relationship established in Caucasian populations may not apply to Asian populations. It is also worth noting that in the few cohort studies of PPGL in Asia [21,22,23], the profiles of the most common mutated gene and the recurring *SDHB* variants vary widely by ethnicity even within Asia. In several case reports and case series from Japan, characteristic variants with PPGL have been reported [24,25,26,27,28]; however, there are no comprehensive national studies. We aimed to investigate the prevalence of germline variants in the major susceptibility genes (*MAX*, *SDHB*, *SDHC*, *SDHD*, *TMEM127*, *VHL,* and *RET*) in the Japanese population with PPGL. Furthermore, we evaluated variant classification based on several updated databases and in silico meta-prediction tools and summarised the clinical and genetic features of patients with PPGLs in Japan.

## 2. Results

### 2.1. Clinical Characteristics of the Study Population

Overall, 370 probands were enrolled in the study (166 males and 204 females; mean age: 43.3 years; range: 6–83). Table 1 summarises the clinical characteristics for the whole study cohort and the FS and AS groups. PCC was present in 153 (41.4%) patients, and 31 of these patients had a bilateral PCC. There were 194 (53.0%) patients with extra-adrenal PGL; 79 (21.4%) had head and neck PGL (HNPGL), of which 6 were bilateral HNPGLs; and 116 (31.4%) had abdominal/thoracic PGL (ATPGL), of which 10 were multiple. Twenty-two (5.9%) patients had multifocal tumours, of which 17 had comorbid tumours of PCC and ATPGL, three had both PCC and HNPGL, and two had both HNPGL and ATPGL. Metastatic PPGL was observed in 63 (17.0%) patients. A positive FS presentation was observed in 43 (11.6%) patients in the entire study cohort. The other 327 (88.4%) patients had an AS presentation. The FS group were diagnosed at a younger age than the AS group (36.0 ± 13.9 vs. 44.3 ± 15.8 years, *p* = 0.001) and more often had bilateral PCC (25.6% vs. 6.1%), whereas the AS group had a greater proportion of unilateral PCC (35.2% vs. 16.3%).

### 2.2. Classification of Profiled Variants

Of the 370 probands, 63 distinct germline variants were identified in 129 probands, excluding benign (B) or likely benign (LB) variants (Table 2). We assessed the pathogenicity of these variants according to the American College of Medical Genetics (ACMG) and the Association for Molecular Pathology (AMP) guidelines [29] (see Materials and Methods). As a result, 24 variants were classified as pathogenic (P), 29 variants as likely pathogenic (LP), and 9 variants as a variant of uncertain significance (VUS). We found nine novel variants, which were not previously reported or registered in any disease database. Among them, 6/9 variants were classified as pathogenic or likely pathogenic, three variants remained VUS.

### 2.3. Frequency of Germline Variants

In our cohort of 370 probands, 120 subjects were found to harbour P/LP variants, thus the prevalence rate of germline variants was 32.4% (Figure 1A). The most common variants were *SDHB* detected in 57 probands (accounting for 47.5% of P/LP variants detected), *SDHD* in 27 probands (22.5%), *VHL* in 18 probands (15.0%), *RET* in 8 probands (6.7%), *MAX* in 5 probands (4.2%), *TMEM127* in 3 probands (2.5%), and *SDHC* in 2 probands (1.7%). Among the 327 AS presentations, 81 probands had P/LP variants, with a prevalence of the germline variant of 24.8% (Figure 1B). Compared to the AS presentation, the FS presentation had a higher percentage of *SDHD* (30.2% vs. 4.3%) and *VHL* (27.9% vs. 1.8%) variants (Appendix A). Almost half (29/63, 46.0%) of the probands who developed metastatic PPGL carried a P/LP germline variant (Figure 1C). Metastatic PPGL were mostly by *SDHB* variant (21/29, 72.4%), followed by *SDHD*, *RET*, *VHL*, and *MAX*.

The frequency of germline variants according to tumour location is shown in Figure 1D (see Appendix A for details). Among the 119 unilateral PCC patients, 11 (9.2%) had the germline variants in either 5 genes (*SDHB*: *n* = 2, *SDHD*: 1, *VHL*: 3, *RET*: 3, *MAX*: 2). In contrast, the variant rate was remarkably high in bilateral PCC (20/29, 69.0%), with *VHL* (10/20, 50.0%) being the most frequently mutated gene. In 77 HNPGL patients, 51.9% (40/77) of them had germline variants. The most frequently mutated genes were *SDHD* (22/40, 55.0%) and *SDHB* (17/40, 42.5%), with one case of *SDHC* variants. Thirty-three per cent of the ATPGL (37/115) patients had germline variants. The majority of these variants were in the *SDHB* (34/37, 91.9%) gene, with one case each of *SDHD*, *VHL*, and *SDHC* variants also detected. In 21 patients with multifocal PPGL, 57.1% (12/21) of the patients had germline variants; mutated genes including *VHL* (4/12), *SDHB* (3/12), *SDHD* (3/12), *MAX* (1/12), and *TMEM127* (1/12). On the other hand, 9/21 (42.9%) cases of multifocal PPGL, 11/28 (39.3%) cases of bilateral PCC, 1/6 (16.7%) cases of bilateral/multiple HNPGL, and 6/10 (60.0%) cases of multiple ATPGL were variant negative (VUS was excluded).

Appendix A shows the age-based frequencies of the germline variants detected at the diagnosis. The distribution of the number of patients in each age group showed a symmetrical distribution with a peak at 31–50 years. While almost all *VHL* variants (15/18, 83.3%) were diagnosed before the age of 40, *SDHB/SDHD* variant positive patients had a wider age distribution including in older patients. Of the 61 patients diagnosed with PPGL even after the age of 60, 25% (15/61) were found to harbour P/LP variants.

### 2.4. Clinical Characteristics of Probands with P/LP Variants

Probands with a P/LP variant were younger at the age of diagnosis than the variant negative group (38.2. ± 15.2 vs. 46.0 ± 15.6 years, *p* < 0.001; Table 3). The variant-negative group had a significantly greater proportion of unilateral PCC compared to the P/LP positive group (108/241 (44.8%) vs. 11/120 (9.2%), *p* < 0.00278). In contrast, the ratio of bilateral PCC to single HNPGL was significantly higher in the P/LP variant-positive group (20/120 (16.7%) vs. 9/241 (3.7%), *p* < 0.00278). The P/LP variant group were more likely to develop metastatic PPGL than the variant negative group (24.2% vs. 13.4%, *p* = 0.029).

### 2.5. Genetic and Clinical Characteristics by Specific Susceptibility Genes

#### 2.5.1. SDHB Variants

Fifty-seven *SDHB* variant-positive probands had 15 distinct variants classified as P/LP (missense: 5, nonsense: 4, splice site variant: 2, small deletion: 1, deletion-insertion across the intron/exon border: 1, large deletions: 2; Table 2). A heterozygous duplication of exon 1 in single probands was classified as VUS. The missense variant c.137G>A (p.Arg46Gln) and the frameshift deletion c.470delT (p.Leu157Ter) were the two most frequent variants occurring in 14/57 (24.6%) probands and 13/57 (22.8%) probands, respectively. Most *SDHB* variant-positive patients belonged to the AS presentation (48/57, 84.2%; Table 3). Of the 57 patients, 54 (94.7%) had PGL (single HNPGL: 17, single ATPGL: 30, multiple ATPGL: 4, multifocal of PCC/ATPGL: 3), and only 3 (5.3%) patients had PCC. The *SDHB* variants group had a significantly greater proportion of metastatic PPGL compared with a variant negative group (14.3% vs. 35.8%, *p* < 0.00625 after Bonferroni correction).

We also investigated associations between *SDHB* variant type (truncating or missense) and patient phenotype (Appendix A). There were no significant differences in age at diagnosis, tumour size, or frequency of metastatic PPGL between the truncating and missense variants. We only noticed that truncated variants of the *SDHB* gene tended to show high frequencies of HNPGL (14/33, 42.4%) as well as ATPGL (17/33, 51.5%).

#### 2.5.2. SDHD Variants

We identified 14 distinct P/LP variants and three VUS in the *SDHD* gene (Table 2). Interestingly, three deletion variants of 10–15 bases were found in exon 3 of the *SDHD* gene, among which, c.285_296del was found in several families with distant origins. We found 13/27 (48.1%) *SDHD* variant carriers with a family history (Appendix A). Among them, nine had a family history of PPGL on the paternal side, the remaining four had a brother/sister or son/daughter with PPGL, and none had a family history on the maternal side. The majority of probands with *SDHD* variants presented with HNPGL or HNPGL combined multifocal tumours (25/27 patients, 92.6%). Eleven per cent (3/27) of patients with *SDHD* variants were metastatic forms, comparable to those in the variants negative group (13.4%, 32/241). There were no significant differences in age at diagnosis, tumour size, or frequency of metastatic PPGL between the truncating (*n* = 9) and missense variants (*n* = 13) (Appendix A).

#### 2.5.3. VHL Variants

The P/LP variants of *VHL* in 18 probands, consisted of 12 missense and a deleterious synonymous variant (c.414A>G, p.Pro138Pro). The p.Pro138Pro has been reported to confer PCC susceptibility by promoting exon 2 skipping and consequently repressing the expression of the full-length *VHL* transcript [32]. The three probands with syndromic presentations had hemangioblastoma, pancreatic cysts, and pancreatic endocrine tumours, respectively. The mean age of *VHL* variant-positive probands was 27.1 ± 13.8 (range: 10–64) years. Patients with *VHL* variants were younger than those in the variant negative group (Table 3). The tumours were bilateral PCC in 10 patients (55.6%), and four (22.2%) had the multifocal disease (PCC with ATPGL). Only one case showed distant metastasis.

#### 2.5.4. Minor Variants Genes

The four *RET* missense variants in eight probands were located at the hot-spot codons 631 and 634 in exon 11. Of the five patients with FS presentations, three had a family history of PPGL, and two had prior medullary thyroid carcinoma. The tumours with *RET* patients were always adrenal and often bilateral (5/8, 62.5%). Three of the eight patients had been diagnosed with metastatic PCC. Five distinct *MAX* variants, four *TMEM127* variants, and two *SDHC* variants were detected (including 3 VUS; See Table 2). Bilateral PCC was often observed as a phenotype of the variants in *MAX* or *TMEM127* genes. Of the two patients with the *SDHC* variant, one had multiple HNPGLs, the other had a single ATPGL.

## 3. Discussion

The present study revealed five major findings. First, in previously unstudied Japanese patients with PPGL, we comprehensively profiled the P/LP germline variants in 32.4% of the total and 24.8% of the AS presentations. Second, heritable PPGL could not be ruled out even at an older age of diagnosis (60–70 years), and the prevalence of germline variants was high (32.2–67.8%) in all tumour locations except unilateral PCC (9.2%). Third, as in the previous reports in European populations, the most frequently mutated gene was *SDHB* (47.5%). However, several *SDHB* variants common in the probands of distinct Japanese families differed from those in other races. Fourth, the incidence of metastatic PPGL in P/LP variant carriers was high (24.2%), especially in *SDHB* carriers (36.8%). Finally, we described nine novel variants, six of which were classified as P/LP.

The overall frequency of germline variants in our study cohort (32.4%) is consistent with that in the reported European cohorts (27.4–32.9%) [5,6,7] and with recent reports in Asians (32.6–34.1%) [22,23,31]. The prevalence of variants in AS presentations vary widely (11.0–36.6%) [6,11,12,18,21,22,23] depending on the definition of “sporadic” and the number of genes investigated. Applying the simple criterion of non-familial and non-syndromic presentation as AS presentation, we found a 24.8% frequency of germline variants in anyof the seven major susceptibility genes (*MAX*, *SDHB*, *SDHC*, *SDHD*, *TMEM127*, *VHL*, and *RET*). Our findings also established that even patients with PPGL diagnosed over the age of 60 may harbour germline variants such as *SDHB* and *SDHD* genes (Appendix A). It should be noted that false-negative family history in patients with PPGL can occur due to patients’ lack of awareness of their relatives’ medical history, the presence of multiple genes with low penetrance in hereditary PPGL [33], and the possibility of transmission by maternal imprinting, which is characteristic of *SDHD*-related PPGL [34]. Nevertheless, the present study performed in the one of the scanty large cohorts in Asia shows that more than one in four PPGL patients had a germline variant, supporting the recommendation for genetic testing in all patients with PPGLs, regardless of age at presentation and family history [2]. Early detection of predisposing germline variants is an important step to detect potentially variant carriers in relatives, as well as the potential for improved outcomes through surveillance of *SDHB* and *VHL* variant carriers [35].

Our cohort confirms the high prevalence of heritability in all tumour locations except unilateral PCC (Figure 1). Of note, bilateral PCC (69.0%) and multifocal PPGL (57.1%) had the strongest association with the presence of a germline variant. These prevalence data are in line with almost all major studies showing that 56.3–90.0% of patients with bilateral PCC and 68.8–85.7% patients with multifocal PPGL have a germline variant [6,18,19,22,23]. In these strongly suspected hereditary bilateral PCC or multifocal PPGL, the most commonly mutated gene was *VHL*, consistent with the priority set by the genetic testing decision algorithm [6,23]. Conversely, unilateral PCC was characterized by a 9.2% frequency of variants, notably lower than that in other tumour locations. From a cost-effectiveness perspective, the value of genetic testing in patients with unilateral PCC lacking symptomatic or metastatic features and without positive family history has not been established [14]. However, in a seminal study, Sbardella et al. showed the usefulness of routine genetic screening with multi-gene panels in PCC patients with genetic heterogeneity [36]. Furthermore, NGS has been recently validated to reduce the processing time and cost compared to conventional Sanger sequencing for each exon.

The detailed breakdown of the genes found to be mutated in our study was unique compared to the non-Japanese cohort studies. Our study showed that *SDHB* was the most mutated gene (47.5% of patients with positive variant and 15.4% of all cases studied), followed by *SDHD* (22.5%, 7.3% of all). While a recent study has suggested a Sino-European difference in the frequency of *SDHB* variants with a lower frequency in East Asian than in Caucasian populations [20], there are other reports which suggest that *SDHB* is the most common variant in Asia [21,22]. These inconsistent results might be attributed to both the differences in criteria for patient selection and the geographic origins of the studies. The present study has the advantage of having a relatively higher number of patients with functional or non-functional HNPGL (79 patients, 21.4% of all PPGL) than other Asian cohort studies (9–39 patients, 2.9–38.6%) [21,22,23]. We found 22/77 (28.6%) of the subjects with HNPGL had *SDHD* variants, and 17/77 (220.1%) had *SDHB* variants (excluding VUS, see Figure 1, Appendix A). These results confirmed previous reports of a high prevalence of *SDHD* variants in HNPGL [37,38] and showed that *SDHB* variants were present at almost comparable frequency. Notably, the truncating variants of the *SDHB* gene showed a high frequency of presentation of HNPGL as well as ATPGL (Appendix A). Further extensive cohort studies and molecular genetics research are required to clarify the association between HNPGL and truncated variants of *SDHB*.

In our cohorts, *SDHB* variants were the most frequently associated variants with distant metastasis occurring in 21/63 (33.3%; Figure 1C). Furthermore, the *SDHB* variant group showed a significantly higher percentage of metastatic PPGL than the variant-negative group (Table 3). These results have been depicted across almost major studies with *SDHB* variants associated with an increased risk of distant metastasis [5,39,40,41]. Of note, there are several recurrent variants among the 16 distinct *SDHB* variants in this study. c.137G>A was the most frequent variant (14/59, 23.7%), followed by c.470delT, which was the second most frequent (13/59, 22.0%; Table 2). In a large UK cohort reported in 2018, *SDHB* intragenic variants were detected in 237 probands, of which c.137G>A was found in 22 cases (9.3%), and this variant was the second most frequent variant in *SDHB* [17]. In the most recent study in the Chinese population, c.137G>A was found in 2/46 (4.3%) of the *SDHB* variant-positive patients, whereas c.136C>T, the most recurrent *SDHB* variant in China (6/46, 13.0%), was not found in our Japanese PPGL population [22]. Inversely, c.470delT, which was reported as a novel variant associated with metastatic PGL in Japan in 2009 [28], has not been profiled in any other recent cohort studies. Although haplotype analysis is required for conclusive determination, a founder effect is highly suspected for c.470delT because this variant is found only in a limited ethnic population. The accumulation and sharing of the phenotypical knowledge (such as the frequency of metastasis and tumour localization) of recurrent variants in each population may lead to the development of variant-specific personalised disease-risk management.

The detection of vast numbers of genetic variants with the recent evolution of sequencing technology in cancer testing has highlighted the importance of standardizing the interpretation and classification of variants among laboratories. Internationally recognised guidelines have been reported by the ACMG and the AMP [29], providing a five-tier classification system based on the combination of multiple lines of evidence with variable rank. In silico prediction of pathogenicity is one of the evidence categories recommended by the ACMG/AMP guidelines. Recently, several in silico meta-prediction tools have been developed based on the analysis of multiple individual scores. Among them, Rare Exome Variant Ensemble Learner (REVEL) has been confirmed to be an excellent predictive tool for assessing the pathogenicity of missense variants [42,43,44]. We used this reliable in silico tool and the latest disease and population databases to facilitate the identification of disease-causing variants. We also profiled nine VUS, including novel variants. Although VUS is excluded from the analysis of the relationship between genotype and phenotype, it is necessary to repeatedly follow the latest database because the class may change by the re-evaluation of VUS by PPGL experts [45].

The present study is the first comprehensive study profiling the prevalence of germline variants in Japanese subjects with PPGL. However, it is essential to acknowledge the study’s limitations. First, due to financial constraints, we could analyse only seven genes using the Sanger sequencing technique. Therefore, it remains possible to carry another minor susceptibility gene, especially in variant-negative multiple PPGLs. However, these seven genes are reported to account for the majority of the germline variants in PPGL [9,45] and exhaust the list of inherited PPGL genes for which secondary findings are required to be reported in the latest ACMG statement (with the exception of the extremely rare *SDHAF2* gene) [46]. Second, this study has the potential for selection bias, thereby falsely increasing the frequencies of germline variants. However, this bias is minimised by the fact that our cohort included more than 50% of subjects with only isolated unilateral benign PCC or ATPGL. Third, we were not able to mention the relationship between the catecholamine profile or SDHB-negative immunohistochemistry and the variant status. This was due to variations in the ability to investigate PPGL systematically among the facilities which provided the samples. Finally, the variant classification to interpret pathogenicity is incomplete because we were not able to perform genetic tests on both parents in all subjects, thereby preventing the determination of the de novo nature of the annotated variants. However, we were able to enhance the accuracy of pathogenicity assessment by reviewing the allele frequencies in multiple population databases, up-to-date disease databases, and utilizing in silico prediction tools.

## 4. Materials and Methods

### 4.1. Patients

The study subjects included Japanese probands clinically and pathologically diagnosed with PPGL referred to the University of Tsukuba Hospital seeking genetic testing during the February 2007–March 2020 period. Genetic testing was performed on patients older than 16 years of age for ethical reasons, although patients diagnosed before the age of 16 years were included. Blood samples and clinical information, including sex, age at diagnosis, tumour location, and extra-paraganglionic metastases, were collected. The study was conducted in accordance with the Declaration of Helsinki and Ethical Guidelines for Human Genome/Gene Analysis Research of Japan. Written informed consent was obtained from all patients. We offered genetic counselling before and after the genetic test, as appropriate.

FS (familial/syndromic) presentation was characterised by the presence of clinical features of multiple endocrine neoplasia type 2 (MEN2) or Von Hippel–Lindau (VHL) disease in the proband or their family members (syndromic) or history of PPGL in the family members (familial), or in combination. The absence of FS features was considered to be AS (apparently sporadic) presentation. Multifocal PPGL was defined as the coexistence of adrenal PCC and PGL or the presence of PGL across multiple areas of the head and neck and abdomen/thorax. Metastatic PPGL was defined by the presence of extra-paraganglionic metastases, including in the lung, liver, bone, and lymph nodes.

### 4.2. Genetic Analysis

Genomic DNA samples were extracted from 10 mL of ethylenediaminetetraacetic acid (EDTA)-treated peripheral blood samples using the Nucleosopin Blood Mini Kit (MACHEREY–NAGEL GmbH & Co. KG, Düren, Germany) according to the manufacturer’s instructions. The following seven genes were analysed: *MAX* (NM_002382.5), *SDHB* (NM_003000.3), *SDHC* (NM_003001.5), *SDHD* (NM_003002.4), *TMEM127* (NM_017849.4), *VHL* (NM_000551.3), and exons 10, 11 and 13–16 of the *RET* (NM_020630.4) proto-oncogene. A priority order was used for genetic analysis according to previous recommendations [5,6]. For instance, *VHL* and *RET* genes were analysed with priority for bilateral PCC, *SDHB* gene for retroperitoneum PGL or metastatic PCC/PGL, and *SDHD* gene for head and neck PGL. When a pathogenic variant was found in one of these genes, no further testing was performed in the remaining genes. After performing PCR using gel electrophoresis for each primer pair, Sanger sequencing was outsourced to a commercial service provider (Eurofins Genomics, Ota-ku, Tokyo, Japan) to detect variants, and the results were analysed at our department. The primers used for the PCR amplification are listed in Appendix A. When no P/LP variant was found by the Sanger sequencing, all exons of the *SDHB*, *SDHC*, and *SDHD* gene were reanalysed using multiplex ligation-dependent probe amplification (MLPA) to examine copy number variation. The MLPA assay was carried out by a commercial service provider (FALCO Biosystems Ltd., Kumiyama, Kyoto, Japan). The sequence results were analysed using Sequence Scanner Software v2.0 (Thermo Fisher Scientific, Waltham, MA, USA) and CLC Sequence Viewer 8.0 software (QIAGEN, Aarhus C, Denmark).

### 4.3. Variant Classification

ACMG/AMP guidelines were used to classify variant pathogenicity [29]. The detected variations were assessed for pathogenic potential using the following in silico tools: Rare Exome Variant Ensemble Learner (REVEL) (https://sites.google.com/site/revelgenomics/downloads, accessed on 21 May 2021), and Human Splicing Finder (https://www.genomnis.com/access-hsf, accessed on 21 May 2021). REVEL is a recently developed in silico variant meta-predictor, which scores rare missense variants on a scale ranging from 0 to 1, with higher scores indicating a greater likelihood of being a disease-causing variant [44]. While REVEL does not suggest a strict threshold for variant categorisation, a score above 0.5 was used for supporting pathogenic variants (ACMG/AMP codes; PP3) with reference to the previously reported criteria [47]. Human Splicing Finder provides information on changes in scores caused by splice site variants in donor and acceptor sites. A reduction of >10% in the predicted score was used as the pathogenic variant in accordance with the original article [48]. Reported interpretations for known variants were obtained through the disease databases such as Human Gene Mutation Database (HGMD) (accessed on 20 May 2021) (http://www.hgmd.cf.ac.uk/ac/index.php; accessed on 20 May 2021) and ClinVar (https://www.ncbi.nlm.nih.gov/clinvar/; accessed on 20 May 2021). In addition, genome aggregation database (gnomAD v3.1.1 (non-cancer); https://gnomad.broadinstitute.org/, accessed on 10 June 2021) and Japanese Multi Omics Reference Panel (jMorp; Genome variation 8.3K JPN (v20200831); https://jmorp.megabank.tohoku.ac.jp/202102/variants, accessed on 10 June 2021) were used as the population database for the estimation of mutant allele frequency (MAF). Only annotated variants with MAF < 0.01 in the gnomAD v3.1.1 (non-cancer) database was retained. All variants absent, or extremely rare, (MAF ≤ 0.00002) in both gnomAD and jMorp were encoded as PM2 in ACMG/AMP. Frameshift, nonsense, deletion of exon(s), and canonical splicing-site change were grouped as protein-truncating variants (Appendix A).

### 4.4. Statistical Analysis

All data were analysed using the SPSS Statistics Version 26 software (IBM Japan Ltd., Chuo-ku, Tokyo, Japan). Data are reported as mean ± S.D., actual numbers, or percentages. Categorical variables were presented as frequency counts and percentages. Comparison of categorical data was performed using the chi-squared test or Fisher’s exact test as appropriate. For data with normal distributions, between-group differences were analysed using the Student’s t-test or one-way analysis of variance (ANOVA). For statistical comparison of nonnormally distributed data, Kruskal–Wallis tests were used. In all statistical tests, two-sided testing was used, and a *p* value of less than 0.05 was considered statistically significant. For multiple comparisons, Bonferroni’s correction was used to adjust the critical *p*-value.

## 5. Conclusions

In this first large-scale examination of the Japanese PPGL cohort, almost a quarter of patients with apparently sporadic PPGL harboured germline variants. This study reinforces the recommendation in Western guidelines to perform genetic testing for PPGL and genotype-based clinical decision-making in the Japanese population.

## Figures and Tables

**Figure 1 cancers-13-04014-f001:**
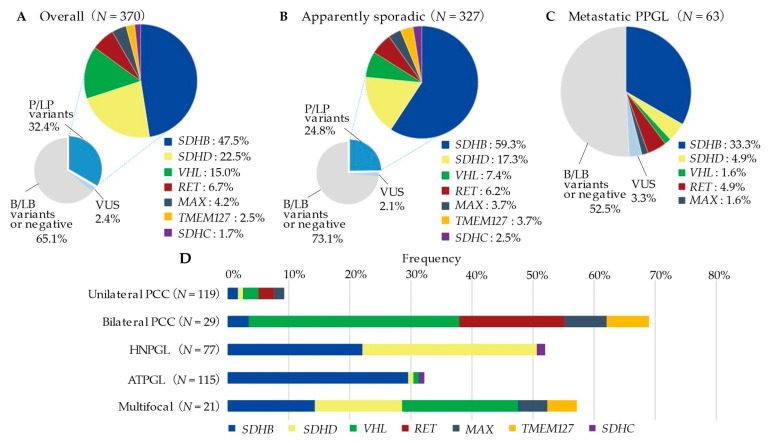
Prevalence and overview of mutated genes by presentation or tumour locations. P/LP, pathogenic/likely pathogenic variants; VUS, variant of uncertain significance; B/LB, benign/likely benign variants; PCC, pheochromocytoma; HNPGL, head and neck paraganglioma; ATPGL, abdominal and thoracic paraganglioma. (**A**) Prevalence and overview of germline variants in the overall PPGL probands (**B**) Prevalence and overview of germline variants in apparently sporadic presentations (**C**) Prevalence and overview of germline variants in the Metastatic PPGL (**D**) Prevalence of P/LP variants by tumour location (VUS was excluded from analysis).

**Table 1 cancers-13-04014-t001:** Clinical characteristics of patients with PPGL in the study.

Characterisic	Overall (*N* = 370)	FS Presentation (*N* = 43, 11.6%)	AS Presentation (*N* = 327, 88.4%)	*p* Value (FS vs. AS)
Sex (M:F)	166:204	25:18	141:186	0.073
Age at diagnosis (years)				0.001
Mean ± S.D.	43.3 ± 15.8	36.0 ± 13.9	44.3 ± 15.8	
Range	6–83	10–69	6–83	
Tumour size (Mean ± S.D.) (cm)	5.4 ± 3.1	4.3 ± 1.6	5.3 ± 3.1	0.184
Location of tumour				<0.001
Unilateral PCC	122 (33.0%)	7 (16.3%)	115 (35.2%)	
Bilateral PCC	31 (8.4%)	11 (25.6%)	20 (6.1%)	
Single HNPGL	73 (19.7%)	9 (20.9%)	64 (19.6%)	
Bilateral/Multiple HNPGL	6 (1.6%)	2 (4.7%)	4 (1.2%)	
Single ATPGL	106 (28.6%)	8 (18.6%)	98 (30.0%)	
Multiple ATPGL	10 (2.7%)	-	10 (3.1%)	
Multifocal (PCC + HNPGL)	3 (0.8%)	-	3 (0.9%)	
Multifocal (PCC + ATPGL)	17 (4.6%)	4 (9.3%)	13 (4.0%)	
Multifocal (HNPGL + ATPGL)	2 (0.5%)	2 (4.7%)	-	
Metastatic	63 (17.0%)	5 (11.6%)	58 (17.7%)	0.392

FS, familial/syndromic; AS, apparently sporadic; PCC, pheochromocytoma; HNPGL, head and neck paraganglioma; ATPGL, abdominal and thoracic paraganglioma.

**Table 2 cancers-13-04014-t002:** List of identified germline variants.

Variant			No. of Probands (FS)	In Silico Prediction		Disease Database		Population Database			ACMG/AMP Class
Gene	Nucleotide Change	Amino Acid Change		REVEL	HSF	HGMD	ClinVar	gnomAD (Global)	gnomAD (East Asian)	jMorp (Japanese)	
*SDHB*	c.79C>T	p.R27*	1	-	-	R	P	1/147,754	1/4946	NR	LP
*SDHB*	c.137G>A	p.R46Q	14 (2)	0.911	-	R	P/LP	1/147,812	0/4966	NR	P
*SDHB*	c.183T>G	p.Y61*	1	-	-	R	P	NR	NR	NR	P
*SDHB*	c.268C>T	p.R90*	2	-	-	R	P	2/147,886	0/4960	NR	P
*SDHB*	c.470delT	p.L157*	13 (2)	-	-	R	P	NR	NR	NR	P
*SDHB*	c.502C>T	p.Q168*	1	-	-	R	P	NR	NR	NR	P
*SDHB*	c.641A>G	p.Q214R	3 (1)	0.973	-	R	US	NR	NR	NR	LP
*SDHB*	c.642G>C	p.Q214H	3 (1)	0.891	-	R	NR	NR	NR	1/16,760	LP
*SDHB*	c.649C>T	p.R217C	1	0.988	-	R	P/LP	1/147,922	1/4950	NR	LP
*SDHB*	c.725G>A	p.R242H	4	0.944	-	R	P	1/147,882	0/4960	NR	LP
*SDHB*	c.201-2A>C	SSC	8	-	−32.0%	NR	LP	1/147,958	1/4968	4/16,760	P
*SDHB*	c.424-2delA ^#^	SSC	2 (1)	-	−91.4%	NR	NR	NR	NR	NR	P
*SDHB*	c.424-7_427 delinsC	SSC and small Indels	2 (1)	-	−75.2%	R	NR	NR	NR	NR	P
*SDHB*	Exon 1 Del	-	1 (1)	-	-	R	P	NR	NR	NR	P
*SDHB*	Exon1 Dup	-	1	-	-	R	US	NR	NR	NR	VUS
*SDHB*	Promotor and Exon1-2 Del	-	1	-	-	R	NR	NR	NR	NR	LP
*SDHD*	c.1A>G	p.M1V	1	0.734	-	R	P	NR	NR	NR	P
*SDHD*	c.3G>A	p.M1I	1 (1)	0.762	-	R	P	NR	NR	NR	P
*SDHD*	c.15G>A	p.W5*	2 (1)	-	-	R	NR	NR	NR	NR	P
*SDHD*	c.57delG	p.L20Cfs*66	1	-	-	R	P	NR	NR	NR	P
*SDHD*	c.112C>T	p.R38*	1	-	-	R	P	NR	NR	NR	P
*SDHD*	c.196T>C ^†^	p.W66R	4	0.936	-	NR	NR	NR	NR	NR	LP
*SDHD*	c.228_239del ^†^	p.L77_L80del	1	-	-	NR	NR	NR	NR	NR	VUS
*SDHD*	c.236T>G ^†^	p.L79R	1	0.968	-	NR	NR	NR	NR	NR	VUS
*SDHD*	c.242C>T	p.P81L	3 (1)	0.908	-	R	P	1/147,924	0/4952	NR	LP
*SDHD*	c.265_279del ^†^	p.S89_Y93del	1	-	-	NR	NR	NR	NR	NR	VUS
*SDHD*	c.285_296del ^†^	p.A96_L99del	5 (3)	-	-	NR	NR	NR	NR	NR	LP
*SDHD*	c.317G>A	p.G106D	3 (3)	0.945	-	R	NR	NR	NR	NR	LP
*SDHD*	c.337_340del	p.D113Mfs*21	2 (1)	-	-	R	P	NR	NR	NR	P
*SDHD*	c.352del	p.D118Mfs*17	1 (1)	-	-	R	P	NR	NR	NR	P
*SDHD*	c.412G>A	p.G138R	1 (1)	0.978	-	NR	LP	NR	NR	NR	LP
*SDHD*	c.315-1G>A	SSC	1 (1)	-	−30.7%	NR	LP	NR	NR	NR	P
*SDHD*	Exon 4 Del	-	1	-	-	R	P	NR	NR	NR	P
*VHL*	c.191G>C	p.R64P	1	0.938	-	R	P/LP	NR	NR	NR	LP
*VHL*	c.235C>G	p.R79G	1	0.792	-	NR	US	NR	NR	NR	VUS
*VHL*	c.250G>A	p.V84M	1	0.758	-	R	LP/US	NR	NR	NR	LP
*VHL*	c.250G>C	p.V84L	1	0.626	-	R	P	1/147,942	0/4948	NR	LP
*VHL*	c.293A>G	p.Y98C	1 (1)	0.938	-	R	P	NR	NR	NR	LP
*VHL*	c.370A>C ^†^	p.T124P	1 (1)	0.88	-	NR	NR	NR	NR	NR	LP
*VHL*	c.371C>T	p.T124I	2 (2)	0.924	-	R	LP	NR	NR	NR	LP
*VHL*	c.407T>G	p.F136C	1	0.821	-	R	P/US	NR	NR	NR	LP
*VHL*	c.414A>G	p.P138=	4 (4)	-		NR	P	NR	NR	NR	P
*VHL*	c.482G>A	p.R161Q	1 (1)	0.797	-	R	P	NR	NR	NR	LP
*VHL*	c.496G>T	p.V166F	1	0.848	-	R	P	NR	NR	NR	LP
*VHL*	c.499C>T	p.R167W	1 (1)	0.868	-	R	P	NR	NR	NR	LP
*VHL*	c.500G>A	p.R167Q	1 (1)	0.874	-	R	P/US	NR	NR	NR	LP
*VHL*	c.524A>G	p.Y175C	1 (1)	0.892	-	R	P/LP	NR	NR	NR	LP
*VHL*	c.548C>T	p.S183L	1	0.620	-	R	US	2/147,912	0/4960	3/16,760	VUS
*RET*	c.1891G>T	p.D631Y	1	-	-	R	P	NR	NR	NR	LP
*RET*	c.1892A>T	p.D631V	1	0.837	-	NR	US	NR	NR	NR	LP
*RET*	c.1900T>C	p.C634R	2	0.972	-	R	P	2/147,640	0/4906	NR	LP
*RET*	c.1901G>A	p.C634Y	4 (3)	0.917	-	R	P	NR	NR	NR	LP
*MAX*	c.3G>A ^$^	p.M1I	1	0.919	-	NR	NR	NR	NR	NR	P
*MAX*	c.70_73del ^†^	p.K24Gfs*40	1	-	-	NR	NR	NR	NR	NR	LP
*MAX*	c.97C>T	p.R33*	1	-	-	R	P	NR	NR	NR	P
*MAX*	c.223C>T	p.R75*	2 (2)	-	-	R	P	NR	NR	NR	P
*MAX*	c.284T>C	p.L95P	1 (1)	0.939	-	NR	US	NR	NR	NR	VUS
*TMEM127*	c.116_119del	p.I41Rfs*39	2	-	-	R	P	3/147,988	0/4964	NR	P
*TMEM127*	c.119C>T	p.S40F	1 (1)	0.715	-	R	US	NR	NR	NR	VUS
*TMEM127*	c.232dupG ^†^	p.D78Gfs*30	1	-	-	NR	NR	NR	NR	NR	LP
*TMEM127*	c.280C>T	p.R94W	1	0.738	-	R	US	9/147,980	0/4960	1/16,760	VUS
*SDHC*	c.43C>T	p.R15*	1	-	-	R	P	3/147,874	0/4964	NR	P
*SDHC*	c.204dupC ^†^	p.I69Hfs*29	1	-	-	NR	NR	NR	NR	NR	LP

REVEL, Rare Exome Variant Ensemble Learner, which scores rare missense variants on a scale ranging from 0 to 1 with higher scores indicating a greater likelihood of that variant being disease-causing, a score above 0.5 was used for supporting pathogenic variants (ACMG/AMP codes; PP3); HSF, Human Splicing Finder, provides information on changes in scores caused by variants in donor and acceptor sites, a reduction of >10% in the predicted score was used as the pathogenic variant; HGMD, Human Gene Mutation Database; Del, deletion (heterozygous); Dup, duplication (heterozygous); SSC, splice site change; R, registered; NR, not registered; P, pathogenic; LP, likely pathogenic; US, unknown significance; VUS, variant of uncertain significance; * termination codon; ^†^ novel variant; ^#^ this variant was described in a previous case report [30]; ^$^ this variant was described in a previous cohort [31].

**Table 3 cancers-13-04014-t003:** Clinical phenotype based on the seven genotypes (excluding VUS).

Characteristic	Variant-Negative (*N* = 241)	P/LP-Positive (*N* = 120)	*p* Value ^#^	*SDHB* (*N* = 57)	*SDHD* (*N* = 27)	*VHL* (*N* = 18)	*RET* (*N* = 8)	*MAX* (*N* = 5)	*TMEM127* (*N* = 3)	*SDHC* (*N* = 2)
Sex (M:F)	102:139	59:61	0.228	23:34	18:9	10:8	4:4	1:4	3:0	0:2
Familial presentation	2	38		9	13	11	3	2	0	0
Syndromic presentation	-	5		-	-	3	2	-	-	
Age at diagnosis (years)			<0.001							
Mean ± S.D.	46.0 ± 15.6	38.2 ± 15.2		37.2 ± 14.5 ^†^	44.6 ± 12.0	27.1 ± 13.8 ^†^	42.1 ± 15.2	39.0 ± 21.1	50.3 ± 10.3	44.5 ± 1.5
Range	6–83	10–74		12–74	23–65	10–64	25–68	13–69	39–64	43–46
Tumour size (cm) (Mean ± S.D.)	5.3 ± 3.1	4.9 ± 2.6		5.5 ± 2.9	3.4 ± 1.2	5.2 ± 1.9	7.1 ± 2.8	4.6 ± 3.9	3.0 ± 0.5	5.4 ± 1.4
Location of tumour			<0.001							
Unilateral PCC	108	11 *		2	1	3	3	2	-	-
Bilateral PCC	9	20 *		1	-	10	5	2	2	-
Single HNPGL	36	35 *		17	18	-	-	-	-	-
Bilateral HNPGL	1	5		-	4	-	-	-	-	1
Single ATPGL	72	33		30	1	1	-	-	-	1
Multiple ATPGL	6	4		4	-	-	-	-	-	-
PCC + HNPGL	2	1		-	1	-	-	-	-	-
PCC + ATPGL	7	9		3	-	4	-	1	1	-
HNPGL + ATPGL	-	2		-	2	-	-	-	-	-
Metastatic	32 (13.4%)	29 (24.2%)	0.029	21 (36.8%) ^‡^	3 (11.1%)	1 (5.6%)	3 (37.5%)	1 (20.0%)	-	-

VUS, variant of uncertain significance; P/LP, pathogenic/likely pathogenic variants; PCC, pheochromocytoma; HNPGL, head and neck paraganglioma; ATPGL, abdominal and thoracic paraganglioma; ^#^
*p* value for the comparison between variant-negative and P/LP-positive group; * analysis performed between variant-negative and P/LP-positive group, *p* value was significant (*p* < 0.00278 after Bonferroni correction) between each tumour location by post hoc tests for the chi-square independence test; ^†^ analysis performed between *SDHB*, *SDHD*, *VHL*, and variant-negative group. Mean value was significantly different *SDHB* or *VHL* vs. variant-negative group by Kruskal–Wallis tests; ^‡^ analysis performed between *SDHB*, *SDHD*, *VHL*, and variant-negative group. *p* value was significant (*p* < 0.00625 after Bonferroni correction) vs. variant-negative group by post hoc tests for the chi-square independence test.

## Data Availability

The authors confirm that the datasets analysed during the current study are available from the corresponding author upon reasonable request.

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
