# Peer review of "Prevalence of Germline Variants in a Large Cohort of Japanese Patients with Pheochromocytoma and/or Paraganglioma"

_cancers, 2021, doi:10.3390/cancers13164014_

Round 1
Reviewer 1 Report
In their manuscript, Yonamine et al describe their results following comprehensive genetic analyses of a Japanese cohort of subjects with PPGL. The data presented aligns with results previously reported in Western/European PPGL cohorts and reinforce the importance of genetic testing in patients who present clinically with PPGL. Overall, the manuscript is well-written, clearly presented and, I believe, will be of interest to the broad readership of Cancers.
Comments:
1) SDHD mutations predominantly follow a maternal mode of transmission. Are the authors able to confirm this is the cases identified in their cohort?
2) In addition to PPGL, a number of other tumours (ie. GIST, RCC) are associated with SDHx mutations. Are the authors able to provide data regarding additional tumour types identified in their cohort of subjects with confirmed SDHx mutations?
3) Was immunohistochemical analysis undertaken for PPGL as part of either standard clinical care or research activity? If so, are the authors able to comment on the association between the conservation of SDHB immunopositivity on IHC with mutation status? For examples, are there subjects in which SDHB immunopositivity was not preserved, however, a mutation in one of the SDHx genes was not identified? Is it possible that such patients have an epimutation in SDHC (or other related gene)? In this regard, a comment in the discussion and appropriate inclusion of the relevant citation(s) reference will be relevant to this manuscript and research study.
Author Response
For reviewer 1
Thank you for reviewing our article despite your busy schedule. Your comments and suggestions have been very helpful. We would like to respond to each of them in this letter.
1) SDHD mutations predominantly follow a maternal mode of transmission. Are the authors able to confirm this is the cases identified in their cohort?
Thanks for your valuable comments. As the SDHD gene is recognized as a maternal imprinted gene, we believe that you meant 'paternal mode of transmission' not 'maternal'. We confirm 9/13 SDHD variant carriers with a family history of PPGL in their paternal lineage. The remaining four probands had a family history of PPGL in a brother/sister or son/daughter. None of the SDHD variant carriers had a family history of PPGL on the maternal side. We added this description to the Results section of the “2.5.2 SDHD variants” (page 7, lines 216–220).
2) In addition to PPGL, a number of other tumours (ie. GIST, RCC) are associated with SDHx mutations. Are the authors able to provide data regarding additional tumour types identified in their cohort of subjects with confirmed SDHx mutations?
Thanks a lot for your comments. As you pointed out, SDHx variants have also been implicated in the genesis of other tumours like clear cell kidney cancer, GIST, and pituitary adenomas. As far as we review the clinical data collected, there were no cases of RCC or GIST preceding PPGL among the probands of the study. We believe this is due to the lower penetration rate of other tumours (RCC up to 14% and GISTs 2% in SDHB carriers) compared to PPGL (Endocr Relat Cancer. 2015.22:T91–103.).
3) Was immunohistochemical analysis undertaken for PPGL as part of either standard clinical care or research activity? If so, are the authors able to comment on the association between the conservation of SDHB immunopositivity on IHC with mutation status? For examples, are there subjects in which SDHB immunopositivity was not preserved, however, a mutation in one of the SDHx genes was not identified? Is it possible that such patients have an epimutation in SDHC (or other related gene)? In this regard, a comment in the discussion and appropriate inclusion of the relevant citation(s) reference will be relevant to this manuscript and research study.
Thanks a lot for your questions on the essential points. Unfortunately, only a limited number of patients in our study were analyzed for SDHB immunostaining due to the different periods and institutions in which they were operated. Therefore, we are not able to comment definitively on the relevance of the variant status and the SDHB immunopositivity. We added that in Limitation (page 10, lines 366–369).
Our co-author Kimura et al. identified 14 (93%) SDHB-negative immunohistochemistry in 15 SDHB-mutated Japanese patients with PPGL (Endocr Relat Cancer. 2014. 21: L13–6.). I In addition, according to Kimura's recent data, all PPGLs from SDHD variants carriers were also negative for SDHB-IHC (unpublished data). Conversely, it is not certain whether all patients who were negative for SDHB-IHC before genetic analysis carry the SDHx gene variant. This is because in some cases genetic analysis has not been offered for a number of reasons, including the high cost of genetic testing.
We agree with the view that if a PPGL patient with SDHB-negative IHC does not have a germline variant of SDHB or SDHD, the possibility of carrying SDHC or another genetic variant should be considered. However, we have not been able to examine the association of SDHB negative IHC with SDHC (or other related genes), and this still needs to be investigated.
Reviewer 2 Report
This is an interesting manuscript that describes the prevalence of germline variants in a large cohort of patients in Japan with pheochromocytoma and paraganglioma. This study included a large population of patients (307). Although some results support previous findings, in these patients, the study is very useful showing the prevalence of germline variants for the first time in Japanese patients. The study was very well-conducted, it is extremely well-written, and all results are carefully and fairly presented. The shortcomings of the present study are also very well outlined.
Critique/suggestions:
- Reference number 3 is only partially correct, and the statement related to this reference needs to be cited better. References related to these tumors as catecholamine-secreting tumors should be used (e.g. Eisenhofer et al.).
- If the authors state that germline variants are present in about 40% of these tumors, again, appropriate articles – original studies and not review should could be cited – e.g. Fishbein et al. Cancer Cell 2017.
- Metastatic potential in these tumors properly associated to their genetic landscape must be well supported by original studies (e.g. Benn et al JCEM, Amar et al. JCO, King et al JCO, Turkova et al. Endocrine Practice etc.).
- What are the ethical reasons that patients younger than 16 years cannot be genetically tested in Japan?
- How many patients with multiple PCCs or PGLs were found to have negative genetic testing? This information could be included in the revised manuscript.
Minor: “analysed” would be “analyzed
Author Response
For reviewer 2
Thank you for reviewing our article despite your busy schedule. Your comments and suggestions have been very helpful. We would like to respond to each of them in this letter.
1) Reference number 3 is only partially correct, and the statement related to this reference needs to be cited better. References related to these tumors as catecholamine-secreting tumors should be used (e.g. Eisenhofer et al.).
Thanks a lot for your comments. According to your suggestion, we used references related to these tumors as catecholamine-secreting tumors. We have corrected these points in the revised manuscript (page 2, lines 81–83).
2) If the authors state that germline variants are present in about 40% of these tumors, again, appropriate articles – original studies and not review should could be cited – e.g. Fishbein et al. Cancer Cell 2017.
Thanks for your helpful comments. Following your valuable input, we have removed the review from the references on prevalence rate and replaced it with the original articles (including "Fishbein et al. Cancer Cell 2017"). We have also corrected the phrase "up to 40%" to "around 30%" (page 2, lines 88).
3) Metastatic potential in these tumors properly associated to their genetic landscape must be well supported by original studies (e.g. Benn et al JCEM, Amar et al. JCO, King et al JCO, Turkova et al. Endocrine Practice etc.).
The reviewer makes a valid point. A summary of the results on the association between SDHB variants and metastatic PPGL and consistency with the references provided has been added to the revised manuscript (page 9, lines 320-324).
4) What are the ethical reasons that patients younger than 16 years cannot be genetically tested in Japan?
Thanks for your fair questions. In Japan, the disclosure of a positive variant result should be explained at the point when the patient's ability to understand has been established, generally after the age of 16. For this reason, until recently, there were strict criteria for genetic testing younger than 16 years. However, with the increasing demand and adoption of genetic testing, the requirements have recently been relaxed.
5) How many patients with multiple PCCs or PGLs were found to have negative genetic testing? This information could be included in the revised manuscript.
Thanks for this vital perspective. Concerning multiple PPGLs, 9/21 (42.9%) cases of multifocal PPGL, 11/28 (39.3%) cases of bilateral PCC, 1/6 (16.7%) cases of bilateral/multiple HNPGL, and 6/10 (60.0%) cases of multiple ATPGL were variant negative (VUS was excluded). We added these descriptions to the results section “2.3. Frequency of germline variants” (page 5, lines 172-174). In addition, the possibility that these variant-negative multiple PPGLs carry another minor susceptibility gene was mentioned in Limitation(page 10, lines 358-359).
Minor: “analysed” would be “analyzed
For consistency with UK English, we use "analyse" as well as "metastasise" and "recognise". We hope you will agree.